# Effects of Alkyl-Substituted Polybenzoxazines on Tribological Properties of Non-Asbestos Composite Friction Materials

**DOI:** 10.3390/polym13040567

**Published:** 2021-02-14

**Authors:** Anun Wongpayakyotin, Chanchira Jubsilp, Sunan Tiptipakorn, Phattarin Mora, Christopher W. Bielawski, Sarawut Rimdusit

**Affiliations:** 1Research Unit on Polymeric Materials for Medical Practice Devices, Department of Chemical Engineering, Faculty of Engineering, Chulalongkorn University, Bangkok 10330, Thailand; anun.wong@hotmail.com (A.W.); phattarin.m@gmail.com (P.M.); 2Department of Chemical Engineering, Faculty of Engineering, Srinakharinwirot University, Nakhonnayok 26120, Thailand; chanchira@g.swu.ac.th; 3Department of Chemistry, Faculty of Liberal Arts and Science, Kamphaengsaen Campus, Kasetsart University, Nakhon Pathom 73140, Thailand; faassntk@ku.ac.th; 4Center for Multidimensional Carbon Materials (CMCM), Institute for Basic Science (IBS), Ulsan 44919, Korea; bielawski@unist.ac.kr; 5Department of Chemistry, Ulsan National Institute of Science and Technology (UNIST), Ulsan 44919, Korea

**Keywords:** brake friction materials, friction and wear, polymer-matrix composites (PMCs), thermosetting resins

## Abstract

A series of substituted polybenzoxazines was synthesized and studied as binders in non-asbestos friction composite materials. The structures of the polybenzoxazines were varied in a systemic fashion by increasing the number and position of pendant alkyl (methyl) groups and was accomplished using the respective aromatic amines during the polymer synthesis step. By investigating the key thermomechanical and tribological characteristics displayed by the composite materials, the underlying structure-properties relationships were deconvoluted. Composite friction materials with higher thermomechanical and wear resistance properties were obtained from polybenzoxazines with relatively high crosslink densities. In contrast, polybenzoxazines with relatively low crosslink densities afforded composite friction materials with an improved coefficient of friction values and specific wear rates.

## 1. Introduction

Over the past 30 years, developments in the automotive industry have provided vehicles with impressive acceleration and speed capabilities. To manage such motion, brake systems are required to slow or stop a moving vehicle, usually via friction and other energy absorption processes, and remain one of the most important safety components of modern automobiles [1,2]. While non-asbestos organic friction materials are commonly used in automotive brake linings and clutch plates, such materials should satisfy a multitude of requirements, including a stable coefficient of friction, high wear resistance under aggressive operating conditions (such as high speeds, high compressive stress, and temperatures that can range up to 370 °C [3]), the ability to generate low noise while in use, and ideally be prepared in an environmentally friendly manner [4,5]. Recently, high-performance polymer composites are being increasingly favored for friction materials and becoming competition for conventional materials [6,7,8].

Composite friction materials, particularly those that are used in braking applications, generally contain four major components: binders, reinforcement fibers, fillers, and friction modifiers [9,10,11,12,13]. The binders are a major constituent and are used to maintain system integrity so that the composite friction materials can display the desired performance outcome while also maintaining key tribological properties (e.g., coefficient of friction and wear rate) and thermal stability. The reinforcement fibers contribute to composite strength, resistance wear rate, impact and thermal degradation, and other properties of friction materials. Friction modifiers are used to further tune the coefficient of friction exhibited by the composite friction materials. Fillers are added to reduce cost, improve some specific properties, and enhance processability.

Phenolic resins are commonly used as binders for composite friction materials in part because they are available at low cost and possess moderate thermal stability at elevated temperatures, typically up to 300 °C [11]. Moreover, phenolics show high mechanical properties such as hardness, compressive strength, creep resistance, and suitable ingredient wet-out [14,15]. Although phenolics are widely used as binders, the resins still have a number of disadvantages [16]. These resins are sensitive to heat and humidity and in situ polymerization starts slowly even at ambient temperature, resulting in its poor shelf life and other serious drawbacks and limitations [10,17]. For example, cracking is often observed during the resin curing step because ammonia gas and other toxins are emitted as by-products [18]. It is brittle and is not resistant to high temperature, which often results in wear loss and fade under 350 °C for the friction materials [14,19]. In order to overcome these, an alternative polymer based on polybenzoxazine was synthesized in this work and tribo-evaluated to investigate the possibility of replacing the currently used conventional one, i.e., phenolics. Polybenzoxazines are known as an interesting new class of phenolic resin, polymerized by the ring-opening reaction of cyclic benzoxazine resins by thermal cure without the need for a catalyst or curing agent and without eliminating any by-products during thermal polymerization. Moreover, benzoxazine resins could be prepared by the Mannich-like condensation of a phenolic compound, a primary amine, and formaldehyde. The wide molecular design flexibility can be obtained due to the limitless choices of phenolic compounds and primary amines in the synthesis process. The molecular design flexibility of polybenzoxazines has been demonstrated such as in a synthesis of a series of benzoxazine resins with varied arylamine groups [20]. The authors reported that for benzoxazine resins based on m-toluidine and 3,5-xylidine, their thermal degradation temperature (T_d_) at 5% weight loss was reported to be about 350 °C, which is significantly higher than aniline-based polybenzoxazine, i.e., about 315 °C, with no significant effect on the final char yield. The highest glass transition temperature or glass-to-rubber transition, i.e., 238 °C, belonged to 3,5-xylidine-based benzoxazine, while the lowest glass transition was of o-toluidine-based benzoxazine, i.e., 114 °C. In addition, additional amounts of arylamine Mannich bridges and methylene bridges in the polybenzoxazine show improved mechanical properties, including higher crosslink densities and rubbery plateau moduli. Glass-to-rubber transition of polymers often leads to substantial change in stiffness, strength, and wear performance of composite friction materials as reported by Wu et al. (2012) [21]. They have studied the effect of glass-to-rubber transition of thermosetting polymer matrices, i.e., phenolic, polybenzoxazine, phenolic-polybenzoxazine, and phenolic-benzoxazine-nitrile rubber, on the friction and wear characteristics of friction materials. Dynamic mechanical thermal analysis and friction test results revealed that glass-to-rubber transition of thermosetting polymers influenced significantly the friction and wear behavior of the composite materials. There was a significant increasing tendency in friction coefficient and wear rate values for all composites when braking temperatures increased to 200 or 250 °C, accompanying the polymer matrix converted from glassy state to rubbery state.

Therefore, in this study, a systematic series of polybenzoxazines outfitted with different alkyl groups was studied as binder matrices in friction composite materials. The benzoxazine resins were synthesized from bisphenol A and an aromatic amine (aniline, *o*-toluidine, *m*-toluidine, *p*-toluidine, or 3,5-xylidine). The selected amines are structurally similar yet feature a different number of pendant alkyl (methyl) groups and oriented in different positions (regioisomers). It was expected that the number and position of these pendant methyl groups would affect the thermomechanical properties and crosslink densities exhibited by the corresponding resins [20]. Moreover, deconvolution of underlying structure-property relationships may enable the realization of new classes of resins that display high thermal stabilities, high glass transition temperatures, a high coefficient of friction value, and/or high wear resistance characteristics.

## 2. Materials and Methods

### 2.1. Materials

Bisphenol A (AR grade), paraformaldehyde (AR grade), and aniline (AR grade) were obtained from PTT Phenol Co., Ltd. (Bangkok, Thailand), the Merck Company (Darmstadt, Germany), and Panreac Quimica SA Company (Barcelona, Spain), respectively. The other amines, including *o*-toluidine, *m*-toluidine, *p*-toluidine, and 3,5-xylidine (AR grade), were purchased from Loba Chemie PVT. Ltd., Bangkok, Thailand. Aramid pulp was provided by DuPont, Tokyo, Japan. Carbon and glass fibers were purchased from S J Sinthuphun Trading Co., Ltd., Bangkok, Thailand, and Toho Beslon Carbon Co., Ltd. Tokyo, Japan, respectively. Iron fibers, friction modifiers, and fillers were provided by Compact International (1994) Co., Ltd. (Bangkok, Thailand).

### 2.2. Synthesis of Benzoxazine Resins

Using a solventless method that is described in a U.S. patent [22], benzoxazine resins (Scheme 1) were synthesized by mixing bisphenol A, paraformaldehyde, and an aromatic amine in a molar ratio of 1:4:2 at 110 °C for 40 min. The resulting light-yellow solid resin was then ground into a fine powder and stored in a desiccator. The labeling scheme used follows the format BA-# where BA refers to benzoxazine resin and # refers to the aromatic amine used for its synthesis: aniline, a; *o*-toluidine, ot; *m*-toluidine, mt; *p*-toluidine, pt; 3,5-xylidine, 35x. Therefore, for example, a benzoxazine resin synthesized from bisphenol A, paraformaldehyde, and aniline would be labeled as BA-a. Labels prefixed with the letter F (e.g., F-BA-a) refer to resins that were used in tested friction materials.

### 2.3. Preparation of Polybenzoxazine Composites

Polybenzoxazine composite-based friction materials were formulated according to the patented invention by Akebono Brake Industry Co., Ltd. (Tokyo, Japan) [18] with some modifications as summarized in Table 1. Various molding formulations were mixed at 120 °C for 40–45 min using an internal mixer to ensure uniform mixing. The compounds were placed in iron molds with specified dimensions and then heated in a compression molder at 200 °C at a pressure of 15 MPa (using a hydraulic press) for 2 h. The fully cured composites were cooled to room temperature before being tested and characterized.

### 2.4. Characterizations

The degradation temperature (T_d_) and char yield of the composites were determined using a thermogravimetric analyzer (PerkinElmer, model Diamond TG/DTA, Bangkok, Thailand). Approximately 20 mg of each specimen was heated from 30 to 1000 °C at 20 °C/min under an atmosphere of nitrogen. The degradation temperature observed at 5% weight loss and the char yield obtained after heating the specimen to 1000 °C were measured and reported.

A NETZSCH dynamic mechanical analyzer (DMA, Bavaria Germany), model DMA242C, was used to assess the thermomechanical properties displayed by the friction materials. The dimension of each specimen was 50 × 10 × 2.5 mm^3^ (W × L × T). The test was performed using a three-point bending geometry. Strain was applied sinusoidally at 1 Hz. The temperature was scanned from 30 to 300 °C at 2 °C/min under an atmosphere of nitrogen. The storage modulus (E′), loss modulus (E″), and loss tangent or damping curve (tan δ) were then obtained. The glass transition temperature (T_g_) was taken as the maximum point observed on the tan δ curve.

The flexural modulus and flexural strength of the composites were determined according to ASTM D790-M93 using a universal testing machine (Instron, model 5567, Bangkok, Thailand). The dimension of each specimen was 25 × 60 × 3 mm^3^. The test was performed using a three-point bending geometry with a support span of 48 mm and at a crosshead speed of 1.2 mm/min. The flexural modulus and flexural strength tests were determined from the obtained load-displacement curves. The modulus and strength values were reported from the average values obtained after analyzing five specimens.

A ball-on-disk tribometer was used with a tungsten carbide as the ball and the different composite formulations as the disks. The cylindrical ball was 6 mm in diameter, whereas the disk specimens were 30 mm in diameter. The ball was assembled under the arm of the equipment applying a uniform load of 10 N against the surface of the specimen disk, which was rotated at a constant speed of 36.6 cm/s in the air at 25 °C. The total distance was set to 1000 m.

Coefficient of friction and wear rates as a function of temperature of specimens were performed using a DSM constant speed friction testing machine (Shandong Alltour Automotive Co. Ltd., Shandong, China) and at different temperatures (100, 150, 200, 250, 300, and 350 °C) according to a Chinese standardized test (GB5763-2008). The specimen dimensions were 25 × 25 × 7 mm^3^. The weight of each test specimen was recorded after each test period to calculate the wear rate using Equation (1) [23].
(1)$$W_{s}=\left(W_{1}-W_{2}\right)/\left(\rho P\nu t\right)$$
where W_s_ is the specific wear rate (cm^3^/Nm), W_1_ is the weight before the test (g), W_2_ is the weight after the test (g), ρ is the density of the specimen (g/cm^3^), P is the applied normal load (N), ν is the relative sliding velocity (m/s), and t is the experimental time (s).

The worn surface of the specimens was evaluated with a JSM-6510A scanning electron microscopy (SEM) from JEOL Ltd., Tokyo, Japan. using an acceleration voltage of 5 kV. All samples were coated with a thin film of gold to make the surfaces conductive.

## 3. Results and Discussion

### 3.1. Thermomechanical Properties of Polybenzoxazine Composites

Dynamic mechanical analysis is a widely used technique to assess the thermal and mechanical properties displayed by composites and can be related to the coefficient of friction, wear properties, and tribological features of such materials. As summarized in Figure 1, the storage modulus (E′) values measured for a series of polybenzoxazine-based composites tested as a function of temperature ranged between 5.23 and 6.83 GPa when measured in their glassy states. Upon inspection of the data, it was hypothesized that the variation in the E′ values observed may be related to the relative crosslink densities of the constituent polybenzoxazines.

To approximate crosslink density, Equation (2) was employed [24]. As summarized in Table 2, the crosslink densities calculated for the polybenzoxazine composites were in a range of 8.72 and 9.86 mol/cm^3^. The highest crosslink density was calculated for F-BA-35x, which indicated that the corresponding resin (BA-35x) reduced polymer segmental motion to the greatest extent among the series tested. Since the aromatic amine (3,5-xylidine) used to synthesize the resin features two (2) methyl groups, it is relatively electron-rich (pKa of its conjugate acid ~4.8) and thus should promote the corresponding polymerization reaction, including the formation of bridges between polymer chains [20]. In contrast, the lowest crosslink density was calculated for F-BA-ot. The aromatic amine (o-toluidine) used to synthesize this resin features a single methyl that is adjacent to the amino group. As such, steric interaction may reduce the intrinsic reactivity of the starting material (pKa of its conjugate acid ~4.4) and attenuate the polymerization reaction. The crosslink densities for the remaining resins roughly correlate with the predicted reactivities of the corresponding amines (pKas of the conjugate acids: p-toluidine, ~5.1; m-toluidine, ~4.7; aniline, ~4.6) and thus provide further support for the hypothesis.
(2)$$log\frac{E_{e}^{’}}{3}=7.0+294\rho_{x}$$
where *E*′*_e_* (dyne/cm^2^) is the equilibrium tensile storage modulus measured in the rubbery plateau and *ρ_x_* (mol/cm^3^) is the crosslink density, which is defined as the molar number of network chains per unit volume.

The glass transition temperature (T_g_) is the temperature at which glassy polymeric materials transition to rubbery states. Such types of thermal transitions can affect the wear performance of composite friction materials [21]. The glass-to-rubber transition temperatures of the polybenzoxazine composites were determined from the maxima of the tan δ curves (Figure 2) that were recorded for each material and ranged between 185 and 260 °C. In some cases, the T_g_ values were measured to be higher than those reported for various types of phenolic composites (224–250 °C) [21,25]. Regardless, inspection of the thermal data revealed that the measured T_g_ values correlated with the calculated crosslink densities and further supported the hypothesis that the reactivity of the aromatic amine starting material governed resin formation.

### 3.2. Thermal Stability of Polybenzoxazine Composites

Thermal stability is an important criterion for evaluating the performance displayed by composite friction materials. As shown in Figure 3, three main degradation steps were identified in the thermogravimetric thermograms recorded for the polybenzoxazine composites. The first step was found to occur over the range of 350–500 °C and attributed to the degradation of the cashew dust and nitrile butadiene rubber (NBR) powder additives as well as the polybenzoxazine. The onset of the second step ranged between ca. 500 to ca. 600 °C and assigned to the degradation of the aramid pulp additive. The final step occurred between approximately 600 and 900 °C, respectively, and was assigned to the degradation of the barium sulfate additive [26].

The temperature observed at five percent weight loss (T_d5_) was also recorded for the polybenzoxazine composites. The measured values ranged from 401 to 414 °C and were found to correlate with the calculated crosslink densities. For example, the highest value belonged to F-BA-35x, and the lowest value was measured for F-BA-ot, in agreement with the trends observed in determining the storage modulus and glass transition temperatures. The residual weight values recorded at 1000 °C for each composite were relatively consistent and ranged between 69–71% (Table 3). In comparison, the T_d5_ and residual weight values measured for the polybenzoxazine composites were higher than those of commercialized phenolic friction composites, i.e., 402 °C and 63%, respectively, that were obtained from experimental.

### 3.3. Flexural Property of Polybenzoxazine Composites

Key mechanical properties, including flexural strength and flexural modulus, have been reported to affect the tribological characteristics displayed by composite friction materials [27]. As shown in Figure 4, the flexural strength values measured for the polybenzoxazine composites ranged between 46.1 and 68.5 MPa, whereas the flexural modulus values ranged between 6.6 and 15.1 GPa. Inspection of the data revealed that the measured flexural values positively correlated with the corresponding crosslink densities [20]. Moreover, the flexural strength and flexural modulus properties of the polybenzoxazine composites were higher in value than the normal range reported for friction materials, i.e., a flexural strength of 10.0–40.0 MPa and flexural modulus of 3.0–8.0 GPa [28].

### 3.4. Tribological Properties of Polybenzoxazine Composites

Tribological properties, which are key parameters when considering friction materials, have been evaluated through analysis of their respective coefficient of friction (COF) values and wear rates [9]. Relationships between the COF and sliding distance of polybenzoxazine composites at room temperature (25 °C) are shown in Figure 5. Average COF values were calculated by comparing the data obtained over the sliding distance of 200 m to 1000 m [26] and ranged between 0.25 and 0.36. The COF was found to be inversely proportional to the storage modulus of the corresponding polybenzoxazine composites tested at room temperature, which may reflect the relative rigidity of the materials [21].

In addition, as shown in Figure 6, the wear rate measured for the polybenzoxazine composites ranged from 2.13 × 10^−7^ to 0.62 × 10^−7^ cm^3^/N·m. The wear rate appeared to decrease in proportion to polymer crosslink density, which may be as expected since higher crosslink density values should promote interaction between the various components and result in a material that is relatively robust. The coefficient of friction and wear rate values of all polybenzoxazine composites were related to worn surfaces after tribological testing at 25 °C of the composites, as presented in Figure 7. It can be seen that the friction films tended to form for the composite based on polybenzoxazine having a higher storage modulus value. This characteristic resulted in a decrease in the coefficient of friction. In addition, as expected for the composite based on polybenzoxazine having a higher storage modulus value (more rigid surface), the smoother worn surface or no removal surface agreeing with the lower wear rate was also observed [29,30].

### 3.5. Tribological Properties of Polybenzoxazine Composites at 100–300 °C

The coefficient of friction as a function of temperature in a range of 100 to 350 °C of the polybenzoxazine composites used in brake pad applications was investigated using a Chinese standard (Standard ID: GB5763-2008, brake linings for automobiles) that is similar to the Thailand industrial standard (Standard ID: TIS97-2557(2014), brake lining for vehicles) [31]. BA-a was selected as a binder because of balanced properties, i.e., thermal and mechanical properties, and tribological properties at 25 °C of the F-BA-a composite. As shown in Figure 8, the COF values measured for the F-BA-a monotonically decreased as a function of temperature. Close inspection of the data indicated that the COF value was relatively constant when the polymer was in a glassy state, and more significant changes were observed once the polymer matrix transitioned to a rubbery state at 236 °C. Regardless, the temperature-dependent COF values measured for the polybenzoxazine-based composite were relatively stable when compared to data reported for phenolic-based analogs (c.f., 0.40 at 100 °C vs. 0.33 at 300 °C) [32].

The specific wear rates measured for the polybenzoxazine composite derived from BA-a as a function of temperature are shown in Figure 9. The values were gradually increased from 0.09 × 10^−7^ cm^3^/N·m at 100 °C to 0.33 × 10^−7^ cm^3^/N·m at 300 °C and sharply increased to 0.66 × 10^−7^ cm^3^/N·m at 350 °C. However, it can see that the wear rate was still be in the standard requirement that is less than 3.5 × 10^−7^ cm^3^/N·m. Our results showed similar trends with the reported works of Lertwassana et al. [26] and Wang et al. [32]. As noted above, increased wear rates were observed at temperatures that exceeded the glass transition temperature of the F-BA-a, i.e., 236 °C. Moreover, at 300–350 °C, the thermoset underwent decomposition or degradation which impeded its ability to bind the fibers and other components [20] and resulted in a reduced a wear resistance and COF value [33]. Regardless, the F-BA-a composite exhibited a relatively stable COF and a low wear rate even at elevated temperatures, such as those found in braking applications. To put these results into context, the specific wear rates exhibited by the F-BA-a composite were lower than values reported for phenolic analogs the temperature range of 100 °C to 300 °C, (0.43 × 10^−7^ to 1.63 × 10^−7^ cm^3^/N·m [32] and thus highlight the potential utility of polybenzoxazine composites in contemporary braking applications.

## 4. Conclusions

Composite friction materials derived from a series of substituted polybenzoxazines were synthesized and studied. The structures of the polybenzoxazines were systemically varied by increasing the number and position of pendant alkyl (methyl) groups that were envisioned to affect the underlying polymerization chemistry. All of the friction composite materials exhibited high T_g_ (185–260 °C), T_d5_ (401–414 °C), flexural strength (46.1–65.6 MPa), flexural modulus (6.6–15.1 GPa), and the underlying thermomechanical properties correlated with the crosslink density of the corresponding polybenzoxazine. For tribological properties, it was found that the coefficient of friction and wear rate of polybenzoxazine friction composites were found to be inversely correlated with the polybenzoxazine crosslink density of each type. The polybenzoxazine synthesized from aniline as a key starting material (F-BA-a) was selected to measure tribological properties at high temperatures in a range of 100–350 °C due to balanced mechanical properties (e.g., storage modulus related to crosslinking density) and thermal properties (e.g., T_g_ and T_d_). The results showed that there were significant constant coefficient of friction and wear rate values for the composite when testing temperatures increased to 300 °C, accompanying the polymer matrix converted from glassy state to rubbery state. While at 350 °C the coefficient of friction gradually decreased, the wear rate sharply increased about two times the value at 300 °C. However, both the coefficient of friction and wear rate values in this temperature range agreed with the standard requirement of brake pads for a passenger car.

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
