# Peer review of "Effects of Alkyl-Substituted Polybenzoxazines on Tribological Properties of Non-Asbestos Composite Friction Materials"

_polymers, 2021, doi:10.3390/polym13040567_

Round 1
Reviewer 1 Report
- The data in Figure 8 and Figure 9 should have error bars to ensure that the data is true and reliable.
- For the same material, the friction coefficient may be different each time, therefore, the friction experiment should be repeated at least three times.
- What is the basis for the ratio of various raw materials in Table 1? Why is this ratio and not others?
- The manuscript did not study the influence of different temperatures on the mechanical properties of the material. Changes in mechanical properties have a great influence on tribological properties.
- It can be seen from Figure 7 that the roughness of the unweared area is different, whether the different roughness is also the reason for the different tribological properties of materials in this paper.
- In Figure 8, does the wear rate of other materials have the same law?
Author Response
Reviewer 1
- The data in Figure 8 and Figure 9 should have error bars to ensure that the data is true and reliable.
Authors’ Response: Thank you very much for your suggestion.
The experimental of tribological property according to Chinese standardized test (GB 5763-2008 Brake linings for automobiles) has been shown in section 2.4 Characterizations. The standard is similar to Japan Industrial Standard (JIS D 4411: Brake Linings and Pads for Automobiles) and Thailand Industrial Standard (TIS97 2557 (2014) Brake lining for vehicles). In general, two specimens used to measure tribological properties, i.e. coefficient of friction and wear rate for each time. Therefore, the results reported in Figure 8 and Figure 9 was the averaged values.
- For the same material, the friction coefficient may be different each time, therefore, the friction experiment should be repeated at least three times.
Authors’ Response: Thank you very much for your suggestion.
The answer is similar to the question no. 1
- What is the basis for the ratio of various raw materials in Table 1? Why is this ratio and not others?
Authors’ Response: Thank you very much for your comment.
In this work, the ingredients of the polybenzoxazine composites have been listed in Table 1 were formulated by basis on the information in the patented invention by Akebono Brake Industry Co., Ltd. (Tokyo, Japan) [Ref. no. 18]. The similar ingredient ratio was presented in our previous work [Ref. no. 26]. We have provided the details about the patented invention in the manuscript highlighted in green color (Page 3).
- The manuscript did not study the influence of different temperatures on the mechanical properties of the material. Changes in mechanical properties have a great influence on tribological properties.
Authors’ Response: Thank you very much for your comment.
The authors agreed with the reviewer that the different temperature shows the effect on the mechanical properties. However, the mechanical property, i.e. storage modulus of almost of polybenzoxazine friction material showed a gradually decrease with increasing temperature until reach glass transition temperature of each polybenzoxazine type. In addition, the requirement of mechanical properties of the polymer friction composites is flexural strength of 10.0-40.0 MPa and flexural modulus of 3.0-8.0 GPa at room temperature as informed in Ref. no. 28. Both values of our polybenzoxazine friction composites showed an agreement with required flexural strength and modulus as can see in Figure 4 (Page 8). Importantly, in previously published work by Wu et al. [Ref. no. 21], they found that there was a significant increasing tendency in coefficient of friction and wear rate values for all composites when braking temperatures increased their glass transition temperature, accompanying the polymer matrix converted from glassy state to rubbery state.
- It can be seen from Figure 7 that the roughness of the unweared area is different, whether the different roughness is also the reason for the different tribological properties of materials in this paper.
Authors’ Response: Thank you very much for your comment.
In general, the surface of specimen is not polished before testing tribological property. Because stress formation from the polishing the specimen may have an effect on the adhesion of the filler embedded in the polymer matrix, resulting in an incorrect wear mechanism.
- In Figure 8, does the wear rate of other materials have the same law?
Authors’ Response: Thank you very much for your comment.
The most wear rate as a function of temperature of polymer composites was found to increase with increasing temperature. For commercial brake pad based on phenolic resin as a polymer binder, the phenolic showed the degradation temperature in the range of 200-250oC as reported by Cheng et al.* Therefore, the phenolic-based brake pad was easy to wear at the higher temperature than 250oC. While, the degradation temperature of polybenzoxazine is approximately 340-380oC [Ref. no. 20]. In consequence, the wear rate of the friction composites based on polybenzoxazines was still stable in a range temperature of 100-300oC and the wear rate at 350oC (0.66´10-7 cm3/N m) was agreed with the wear rate (less than 3.5´10-7 cm3/N m) of the testing standard for brake pad, i.e. GB 5763-2008: Brake linings for automobiles (China)/ JIS D 4411: Brake Linings and Pads for Automobiles (Japan)/TIS97 2557 (2014) Brake lining for vehicles (Thailand). We realized that the similar tends of polymer friction materials should be cited as highlighted in yellow color (Page 12).
*Cheng, Y., Decomposition of five phenolic compounds in high emperature water. J. Braz. Chem. Soc. 2014, 25, 2102–2107.
Reviewer 2 Report
In this work, the authors synthesized a series of bisphenol-A based benzoxazines and used them to prepare composites for friction applications. The idea is good, and the work shows a good example of industrial application for benzoxazine resins. In addition, the scope of the paper is clear, and experiments were well sought out. Thus, I recommend it publish on this journal with a minor revision because the manuscript need be improved and the specific comments are as follows:
- The chemical structures of benzoxazine monomers should be provided in the main text.
- Figure 2
The values of the loss tangent in the ordinate should be provided.
- Figure 5
Each line in this figure cannot be distinguished very well, so it needs a further modification.
Author Response
Reviewer 2
In this work, the authors synthesized a series of bisphenol-A based benzoxazines and used them to prepare composites for friction applications. The idea is good, and the work shows a good example of industrial application for benzoxazine resins. In addition, the scope of the paper is clear, and experiments were well sought out. Thus, I recommend it publish on this journal with a minor revision because the manuscript need be improved and the specific comments are as follows:
- The chemical structures of benzoxazine monomers should be provided in the main text.
Authors’ Response: Thank you very much for your suggestion.
We have provided the chemical structure of benzoxazine monomers as your suggestion as shown in Scheme 1 (Page 3) in the revision.
- Figure 2, the values of the loss tangent in the ordinate should be provided.
Authors’ Response: Thank you very much for your comments.
In this study, glass transition temperatures of the polybenzoxazine composites were determined from the peak maxima in the loss tangent (tan δ) investigated by DMA. To ease comparison, the results was plotted and presented in stacking lines. The height of loss tangent would not be compared in this case.
- Figure 5, each line in this figure cannot be distinguished very well, so it needs a further modification.
Authors’ Response: Thank you very much for your suggestion.
We have modified the figure as your suggestion as shown in Figure 3 (Page 7) as well as Figure 5 (Page 9).
Reviewer 3 Report
Reviewers' comments:
Manuscript number: polymers-1090057
Title: Effects of Alkyl-substituted Polybenzoxazines on Tribological Properties of Non-Asbestos Composite Friction Materials.
Comments:
The manuscript reported on Effects of Alkyl-substituted Polybenzoxazines on Tribological Properties of Non-Asbestos Composite Friction Materials. The manuscript needs a detailed editing. It cannot be recommended for publication in the present form. I hope the following points would be helpful for the authors.
(1) Minor English corrections is required throughout the manuscript.
(2) The introduction section should be improved; more related papers must be discussed and superiority, novelty, critical improvement in this study must be clarified.
(3) The figures are too pixelated to be useful. Please provide sharper pictures with better resolution.
(4) 2.3. Preparation of Polybenzoxazine Composites - must be improved.
(5) Figures 3 and 7 – not clear make clear.
(6) Please provides the references for all equations and formula.
(7) In part SEM: how the energy of the accelerator beam used?
(8) Several faults: are added or missing spaces between words: see PDF file.
(9) Conclusions, the author should add some qualitative data of the results.
(10) References: there are recent references in 2019 and 2020 treating the same subject, you can use.
So that I recommended this manuscript to major revision and for future process.
Author Response
Reviewer 3
The manuscript reported on Effects of Alkyl-substituted Polybenzoxazines on Tribological Properties of Non-Asbestos Composite Friction Materials. The manuscript needs a detailed editing. It cannot be recommended for publication in the present form. I hope the following points would be helpful for the authors.
(1) Minor English corrections is required throughout the manuscript.
Authors’ Response: Thank you very much for your comment.
English corrections throughout the manuscript have made as suggested by reviewer.
(2) The introduction section should be improved; more related papers must be discussed and superiority, novelty, critical improvement in this study must be clarified.
Authors’ Response: Thank you very much for your comment.
We realized that the important motivation of the polybenzoxazine based friction materials should be provided. The details about the novelty of this research have been added in the introduction as highlighted in blue color (Page 1 and 2).
(3) The figures are too pixelated to be useful. Please provide sharper pictures with better resolution.
Authors’ Response: Thank you very much for your suggestion.
The resolution of figures, i.e. Figure 3 (Page 7) and Figure 5 (Page 9), were modified to be clear as suggested by the reviewer.
(4) 2.3. Preparation of Polybenzoxazine Composites - must be improved.
Authors’ Response: Thank you very much for your comment.
We have provided more details about the preparation of polybenzoxazine composites as suggested by the reviewer. The revision are highlighted in yellow color (Page 3).
(5) Figures 3 and 7 – not clear make clear.
Authors’ Response: Thank you very much for your suggestion.
We have modified the resolution of both figures as shown in Page 7, and Page 10, respectively.
(6) Please provides the references for all equations and formula.
Authors’ Response: Thank you very much for your suggestion.
We have cited the reference for the equation and formula into the original manuscript highlighted in green color (Page 4 and 5).
(7) In part SEM: how the energy of the accelerator beam used?
Authors’ Response: Thank you very much for your question.
Scanning electron microscopy at acceleration voltage of 5 kV was used to observe the worn surface of the polybenzoxazine friction composites. The information of SEM was in section 2.4 Characterizations (Page 5)
(8) Several faults: are added or missing spaces between words: see PDF file.
Authors’ Response: Thank you very much for your comment.
We have corrected the faults as suggested by the reviewer in the original manuscript already.
(9) Conclusions, the author should add some qualitative data of the results.
Authors’ Response: Thank you very much for your comment.
We realized that we should explain and provide more qualitative finding in the conclusion. Therefore, we have concluded the obtained results into the original manuscript and highlighted with green color. (Page 12)
(10) References: there are recent references in 2019 and 2020 treating the same subject, you can use.
Authors’ Response: Thank you very much for your suggestion.
The additional references in 2020 about polymer composites for brake pad application as can see below have been added in the introduction sections (in Page 1 with highlight of blue)
Additional references:
Ref.no. 6 Biczó, R.; Kalácska, G.; Mankovits, T. Micromechanical Model and Thermal Properties of Dry-Friction Hybrid Polymer Composite Clutch Facings. Materials, 2020, 13, 4508. doi.org/10.3390/ma13204508
Ref. no. 7 Padhan, M.; Marathe, U.; Bijwe, J. Surface topography modification, Film transfer and Wear mechanism for fibre reinforced polymer composites—An Overview. Surf. Topogr.: Metrol. Prop. 2020, 8, 043002. doi: 10.1088/2051-672X/abbcb6
Ref. no. 8 Bhatt, B., Kalel, N., Abdel-Latif, M., Bijwe, J., "Influence of Increasing Amount of Attapulgite on the Performance Properties of Cu-Free Brake-Pads," SAE Technical Paper 2020-01-1601, 2020, https://doi.org/10.4271/2020-01-1601.
Round 2
Reviewer 1 Report
Figure 8 and Figure 9 should show the error bar of the curve. What reviewers mean is not that the data in the graph is not average
Reviewer 3 Report
The authors revised the manuscript according to the reviewers' comments.